Microplastic ingestion by commercial marine fish from the seawater of Northwest Peninsular Malaysia

Foo Yuen Hwei 1
Ratnam Sharnietha 2
Lim Er Vin 2
Abdullah Masthurah 1 2
Molenaar Vincent J. 3
Shau Hwai Aileen Tan 1 2
Zhang Shoufeng 4
Li Hongjun 4
Mohd Zanuri Norlaila Binti lailazanuri@usm.my 2
1 School of Biological Sciences, Universiti Sains Malaysia (USM) , Gelugor , Pulau Pinang , Malaysia
2 Centre for Marine and Coastal Studies (CEMACS), Universiti Sains Malaysia (USM) , Gelugor , Pulau Pinang , Malaysia
3 Athena Institute for Research on Innovation and Communication in Health and Life Sciences, VU University Amsterdam , De Boelelaan , Amsterdam , Netherlands
4 National Marine Environmental Monitoring Center , Dalian , China
Bordon Isabella
Electronic publication date: 2022 Apr 19
Publication date: 2022
Volume: 10
Electronic Location ID: e13181
Received 2021 Apr 6; Accepted 2022 Mar 7
Copyright: ©2022 Foo et al.
Copyright year: 2022
Copyright holder: Foo et al.
License: This is an open access article distributed under the terms of the Creative Commons Attribution License, which permits unrestricted use, distribution, reproduction and adaptation in any medium and for any purpose provided that it is properly attributed. For attribution, the original author(s), title, publication source (PeerJ) and either DOI or URL of the article must be cited.
License URL: https://creativecommons.org/licenses/by/4.0/

Keywords: Microplastic ingestion, Commercial marine fish, Northwest Peninsular Malaysia seawater

Funding: The Ministry of Higher Education Malaysia for Fundamental Research Grant Scheme FRGS/1/2020/STG03/USM/02/6 Universiti Sains Malaysia Bridging Incentive Grant project 304/PPANTAI/6316350 This work was supported by the Ministry of Higher Education Malaysia for Fundamental Research Grant Scheme with project code: FRGS/1/2020/STG03/USM/02/6 and Universiti Sains Malaysia, Bridging Incentive Grant project code 304/PPANTAI/6316350. The funders had no role in study design, data collection and analysis, decision to publish, or preparation of the manuscript.

==============================
Over the past decade, concerns over microplastic pollution in the marine ecosystem has increasingly gained more attention, but research investigating the ingestion of microplastics by marine fish in Malaysia is still regrettably lacking. This study investigated the microplastic presence, abundance, and morphological types within the guts of four species of commercial marine fish (Atule mate, Crenimugil seheli, Sardinella fimbriata and Rastrelliger brachysoma) caught in seawater off the coast of Malaysia’s Northwest Peninsular. A total of 72 individual commercial marine fish guts from four species (fish per species n = 18) were examined. Remarkably, this study found that 100% of the samples contained microplastics. A total number of 432 microplastics (size < 5 mm) from the four species were found in the excised marine fish guts. The most common type of microplastic discovered was fragment, which accounted for 49.5% of all microplastics present. The gut microplastic content differed between species. Sardinella fimbriata recorded the greatest amount of microplastic ingestion, with an average microplastic count of 6.5 (±4.3) items per individual fish. However, there were no statistically significant differences found when comparing study species and different locations. SEM-EDX analysis confirmed the presence of microplastic particles by identifying the chemical elements found in the samples. Since the four studied species of commercial marine fish are popular protein sources in Malaysians’ daily diet, this study suggests potential microplastic exposure to humans via contaminated fish consumption in Malaysia, which was previously unknown. Based on previous scientific evidence, this study also demonstrates the high probability of microplastic ingestion in marine fish in the Malaysian seawater, which could have an adverse effect on fish health as well as marine biota.

Introduction

Plastic pollution in the marine ecosystem has sparked increasing interest and research over the past decades, emphasizing the ecological and biological consequences for marine biota (Andrady, 2011). Global plastic production has surged since the early 1950s, reaching 360 million tonnes by 2018. Substantial demand from various sectors is driving this trend, among which domestic usage is an important source (Cole et al., 2011). Plastic is mass-produced on a large scale due to its high durability, resistance to degradation, relative ease of production and low production cost. Plastics are extremely resistant to biodegradation. They do, however, degrade into smaller particles over time when exposed to several natural factors, such as sunlight and wave action (Wang et al., 2016). The increased use of plastics in society has led to an exponential growth in plastic production, which is expected to continue. Plastics will increasingly reach all areas of the environment due to this increase in production and associated mismanagement during production, distribution, use and final disposal (Azoulay et al., 2019; Lusher et al., 2017). Eriksen et al. (2014), based on 24 expeditions, estimated that at least 5.25 trillion buoyant particles, weighing around 268.940 tons, are in the Earth’s oceans.

The National Oceanic and Atmospheric Administration (NOAA) has defined microplastics as tiny plastic fragments smaller than five mm in diameter (Barboza & Gimenez, 2015). This means that most microplastics are difficult to identify with the naked eye, requiring microscopic observation. According to Li, Liu & Paulchen (2018), microplastics can be divided into two major types—primary and secondary microplastics. Primary types of microplastics include moulded plastic powders, ‘scrubbers’ for surface blast cleaning, industrial plastic nanoparticles, and microbeads found in cosmetic products. Also, spherical or cylindrical virgin resin pellets that are usually five mm in diameter are widely used before and during the plastic manufacturing processes (Koehler et al., 2015). Secondary microplastics are formed after the degradation or fragmentation of larger plastic debris (Fok & Cheung, 2015).

Primary and secondary microplastics have the potential to be ingested among a wide range of marine ecosystem taxa, such as benthic organisms, corals, plankton, fish and large marine mammals (Sharma & Chatterjee, 2017). The extremely small size of microplastics (between 0.1 µm and 5 mm in diameter) makes them highly bioavailable. Due to their buoyancy and appealing colours, they can be easily ingested by fish (Jovanović, 2017). Marine fish might mistakenly ingest microplastics, with detrimental effects, as they look like natural prey. Marine fish play a vital role in the marine ecosystem, linking both lower and higher trophic levels, acting as both prey and predator. Ultimately, trophic transfers can occur from lower to higher levels within a food chain, potentially causing relatively greater exposure to microplastics among apex predators (Santillo, Miller & Johnston, 2017).

A recent study by Karami et al. (2017a) in which packets of dried fish (C. subviridis, J. belangerii, R. kanagurta and S. waitei) were purchased from local markets in Malaysia, found that microplastics were present in the edible flesh of these four commercial marine fish species. The authors estimated that 246 microplastic particles from these dried fish sources are consumed annually by humans. Fish is an important natural protein source in the daily diet of many nations, including the majority of Malaysians (Teh, 2012). Although microplastic ingestion by fish has previously been reported worldwide, there is relatively limited information on Malaysia’s commercial coastal species. Based on statistics from 2000, annual per capita fish consumption in Malaysia was 58 kg per person (Nurnadia, Azrina & Amin, 2011). The 2008 Malaysian Adult Nutrition Survey (MANS) found that the daily consumption prevalence of marine fish in Malaysia was 51% for rural adults and 34% for urban adults (Norimahak et al., 2008). Thus, the hypotheses for this study were that plastic ingestion rates do not differ between species of fish and the habitats in which the fish were caught. Therefore, this research is crucial in highlighting the significant consumption of commercial marine fish as a potentially important source of microplastics exposure in Malaysia.

Materials and Methods

Sample Collection

In this study, four commercial marine fish species, Atule mate (Yellowtail scad), Crenimugil seheli (Bluespot mullet), Sardinella fimbriata (Fringescale sardinella) and Rastrelliger brachysoma (Short mackerel) with a total number of 72 fish [fish per species n =18] were attained from the fishermen at Teluk Bahang and Penaga fish market, Penang. The GPS coordinates of the fishing locations were obtained from the fishermen. Crenimugil seheli were collected on November 27th, 2019 at the Teluk Bahang fish market from fishing site GPS coordinates 5°25′38.33″N 100°8′54.23″E (Fig. 1). Atule mate, Sardinella fimbriata and Rastrelliger brachysoma were collected on December 3rd, 2019 at Penaga fish market from fishing location 5°35′30.3″N 100°15′38.0″E. Both fishing sites are located in Northwest Peninsular Malaysia seawater. The animals were kept on ice during transportation to the laboratory at the Centre for Marine and Coastal Studies (CEMACS), Universiti Sains Malaysia.

Figure 1 Map of fishing locations. St 1: fishing location of Crenimugil seheli; St 2: fishing location of Atule mate, Sardinella fimbriata and Rastrelliger brachysoma.

Laboratory Procedures

Laboratory procedures were carried out at the microplastic laboratory, Centre for Marine and Coastal Studies (CEMACS), Universiti Sains Malaysia. Several laboratory steps were conducted in processing the samples before further microplastic observation and identification, including (1) measurement of biometric parameters of each fish sample, (2) isolating, which included steps of (i) depuration, (ii) dissection and (iii) digestion, (3) digestant filtering and (4) density separation and filtering.

Measurement of Biometric Parameters

Biometric parameters of the samples were obtained by measuring their standard lengths, total lengths and total wet weight. The standard length was measured from the fish mouth until tail muscle while the total length was measured from mouth to tail.

Isolating

The isolation process was done by the extraction of microplastics from biotic materials. The process included depuration, dissection and digestion of biological tissues with chemical processes (Lusher et al., 2017).

Depuration

To ensure that the study focused primarily on fish ingestion of microplastics, external adhering (micro)plastics were removed by washing with filtered distilled water. This process was to ensure that only microplastics retained within the tissues or collected in the intestinal tract were included, and findings were not confounded by other adhering microplastics (Lusher et al., 2017).

Dissection

All dissection equipment such as scalpel, dissecting scissors and tweezer were rinsed with filtered distilled water to prevent contamination. A total of 72 fish samples were dissected with a scalpel and dissecting scissors from the anus to the upper part near the gills to obtain the fish guts. The guts were then excised using tweezers. The excised guts were weighed on an electronic balance, recorded and kept in 200 mL clean covered glass jars.

Digestion

Microplastics might be masked as they can be disguised by biological materials through encapsulation by the mucosae. Thus, digestion was done using 10% potassium hydroxide (KOH) as a base to denature protein and hydrolyse chemical bonds. KOH pellet (100 g) was dissolved in distilled filtered water (1,000 mL) to get a 10% KOH solution (Karami et al., 2017a). A total of 5,000 mL of KOH was prepared to digest the four species of fish guts. Upon usage, KOH was filtered through Whatman GF/C glass microfibre filter membrane (pore size: 1.2 µm) to prevent plastic and other debris contamination. 30% volume of 10% KOH (1:3 v/v) was then added to each glass jar containing extracted fish gut (Karami et al., 2017a). One blank sample was prepared and processed simultaneously for each fish species (total four blanks). All the glass jars with covers containing extracted fish guts in 10% KOH solution were left in the incubator at 60 °C for 24 h until the gut digestion was completed (Dehaut et al., 2016).

Digestant filtering

The gut specimens were taken out from the incubator after 24 h of digestion. The presence of a clear digestant showed that the digestion process had been completed. Only guts from Crenimugil seheli obtained clear digestant and proceeded with the direct vacuum filtration process using Whatman GF/C glass microfibre filter membrane (pore size: 1.2 µm). Each filter was kept individually in a clean petri dish for further processing.

For Atule mate, Sardinella fimbriata and Rastrelliger brachysoma, a little digestion-resistance was found where some particles sank at the bottom of the glass jar. A density separation process was performed on these guts for further processing.

Density separation and filtering

Indigestible particles from the guts of Atule mate, Sardinella fimbriata and Rastrelliger brachysoma samples were density separated using potassium iodide (KI) solution. A 1:1 ratio of distilled water to KI was used to obtain a solution density of 1.52 g/mL (adaptation of Karami et al., 2017a). KI was used due to its non-hazardous and high-density properties and to allow less dense microplastic particles to separate from denser inorganic large particles such as fish bones and sand (Bergmann, Gutow & Klages, 2015). A density separator consisting of 200 mL filter funnels, 100–125 mL conical flasks, retort stands, rubber tubes, hinged and binder clips was set up. Filter funnels and conical flasks were covered with aluminium foil to prevent airborne contamination. KI was filtered through Whatman GF/C glass microfibre filter membrane (pore size: 1.2 µm) to prevent contamination. 100 mL of KI was added to each gut sample of Atule mate, Sardinella fimbriata and Rastrelliger brachysoma containing indigestible particles in KOH, which were then poured into the filter funnels. Samples were then allowed to settle for one hour. After the denser materials settled, the rubber tubes were unclipped slowly to discharge settled and unwanted particles. The remaining supernatant was collected in a clean conical flask and unwanted particles were discarded. Supernatant in the conical flasks was filtered through Whatman GF/C, and each filter was kept individually in glass petri dishes for further observation processing.

Microplastic analysis

Visual identification of microplastics on filter membranes was carried out by using an MDSI-40X dissecting stereomicroscope and a DM4 (1000x) USB digital electronic microscope. Microplastics were analysed and classified into different types based on their morphological characteristics, i.e., fibre, pellet, film and fragment (Crawford & Quinn, 2016; Karami et al., 2017a). Significant microplastics found from filters were photographed. The number of microplastics on the filter membranes was then recorded and expressed as items per individual fish. Due to the high susceptibility of error with visual identification, additional physical and characterization on microplastics using SEM-EDX technique was performed to reduce the risk of incorrect interpretation.

SEM-EDX analysis

A Scanning Electron Microscope (SEM) was used to examine the characterization of the surface composition of the microplastic particles. Ten pieces of microplastic samples with different morphology were randomly selected and analyzed using SEM. Several shapes of microplastics were found, including regular sphere, flat fragments, film and fibre. The quality of the microstructure element (chemical characteristics) was assessed using the Energy-Dispersive Spectroscopy (EDX) analysis.

Statistical analysis

All statistical analyses were performed using SPPS (v23). The number of microplastics was log-transformed for analysis. All data were back-transformed for presentation. The total plastic ingestion rates across different species and habitat were determined by analysis of variance (ANOVA) when assumptions for normality and homoscedasticity were met (Shapiro–Wilk and Levene test, respectively). The significance level was set at α = 0.05. Later, the Tukey’s post hoc test was used to identify the differences among species. Data that did not meet normality and homoscedasticity were subjected to non-parametric Kruskal- Wallis tests followed by a Wilcoxon-Mann–Whitney post hoc test. Lastly, to investigate the differences in ingestion rate in two habitats, an independent samples T-test was conducted.

Results

Biometric parameters of fish samples

Table 1 shows the mean biometric parameters (±SD) of the 72 individual fish for four species (n = 18 per species) investigated. Crenimugil seheli had the highest mean total length and standard length, at 16.99(±0.68) cm and 14.32(±0.52) cm respectively. Sardinella fimbriata had the lowest mean total length and standard length at 14.74(±0.91) cm and 12.09(±0.78) cm respectively. Crenimugil seheli was ranked the highest mean wet weight (whole fish) with 45.67(±4.03) g while the lowest mean wet weight species was Sardinella fimbriata with 29.17(± 6.26) g. For extracted fish gut weight, the highest mean weight was attributed to Rastrelliger brachysoma with 2.95(±0.55) g while Sardinella fimbriata was ranked with the lowest mean of 1.28(±0.31) g.

Table 1 Mean biometric parameters (±SD) of four studied fish species (n = 18 per species).

Species	Commonname	Total length (cm)	Standard length (cm)	Wetweight (g)	Gut weight (g)	
Atule mate	Yellowtail scad/Pelata	16.68 (±0.47)	13.02 (±0.55)	40.17 (±5.24)	1.58 (±0.43)	
Crenimugil seheli	Bluespot mullet/Kedera	16.99 (±0.68)	14.32 (±0.52)	45.67 (±4.03)	2.60 (±0.64)	
Sardinella fimbriata	Fringescale sardinella/Tamban	14.74 (±0.91)	12.09 (±0.78)	29.17 (±6.26)	1.28 (±0.31)	
Rastrelliger brachysoma	Short mackerel/ Kembung	16.30 (±0.55)	13.33 (±0.33)	43.61 (±2.93)	2.95 (±0.55)	

Presence, abundance and morphological types of microplastics found

A total of 432 pieces of microplastic (size <5 mm) were observed in 72 excised marine fish guts of four commercial species (n = 18 per species) (Table 2). Among the samples of fish guts examined, microplastics were present in 100% of the samples (FO = 100%, Table 2). Sardinella fimbriata had the highest average microplastic number (abundance), at 6.5(±4.3) items per individual fish. Crenemugil seheli had the lowest average microplastic number, at 5.0(±3.7) items per individual fish.

Table 2 Total number of microplastics (MPs), average microplastic number per individual fish (±SD), and frequency of occurrence of plastic ingestion (FO) per individual fish in four studied species (n = 18 per species).

Fish species	FO (%)	Average MPs/individual fish (±SD)	Total number of MPs found	
Atule Mate	100	6.3 (±4.9)	114	
Crenemugil seheli	100	5.0 (±3.7)	90	
Sardinella fimbriata	100	6.5 (±4.3)	117	
Rastrelliger brachysoma	100	6.2 (±3.3)	111	
Total			432	

There were four morphological types of microplastics found: fragment, fibre, pellet and film. Figure 2 shows that fragment was the most frequent type of microplastic found in this study (49.5%), followed by fibre (41.9%) and pellet (7.6%). The film was the least frequent type of microplastic found (0.9%) overall.

Figure 2 Microplastic classification by the morphological types.

Comparison of the abundance and morphological types of microplastics in four selected commercial marine fish guts

The microplastic abundance in the excised marine fish guts of Crenimugil seheli, Atule mate, Sardinella fimbriata and Rastrelliger brachysoma are shown in Fig. 3. ANOVA indicated no significant difference when all species were considered (F = 0.408, p = 0.748). Figure 4 shows the average microplastic abundance per individual fish according to two different locations. In our study population, Crenimugil seheli caught at location 1 showed fewer microplastic concentrations in the gut content compared to the other species caught at location 2. However, independent sample T-test found no significant different between these two locations (t = 0.804, p = 0.424).

Figure 3 Microplastic abundance (number) in the excised marine fish guts of Crenimugil seheli, Atule mate, Sardinella fimbriata and Rastrelliger brachysoma caught from the two fishing locations.

Figure 4 Average microplastic abundance per individual fish in the excised marine fish guts of Crenimugil seheli, Atule mate, Sardinella fimbriata and Rastrelliger brachysoma caught from the two fishing locations.

Crenimugil seheli was observed with the lowest numbers of microplastics in the gut samples among the 4 commercial marine fish species, which were 90 pieces of microplastics in total (n = 18) or 5.0(±3.7) items per individual on average. Relatively similar abundances were found among the three species collected from location 2, although amounts were slightly greater in Sardinella in which there were 117 pieces of microplastics in total (n = 18) or an average of 6.5(±4.3) items per individual fish. By comparison, there were a total of 111 pieces of microplastic found in Atule mate gut samples (n = 18) or 6.3(±4.9) items per individual fish and 114 pieces of microplastics found in guts of Rastrelliger brachysoma (n = 18) with an average of 6.2(±3.3) items per individual fish.

Figure 5 shows that fragments were the most abundant microplastics found in Sardinella fimbriata and Rastrelliger brachysoma with an average number of 4.3(±3.5) items and 3.9(±2.2) items per individual fish, respectively. For Atule mate and Crenimugil seheli, there were an average of 3.5(±2.5) and 3.0(±2.2) fibres per individual sample fish gut respectively, which was the most abundant morphological type of microplastics found in the two species. For all four fish species, film was the least frequent microplastic type present with maximum average numbers of 0.1(±0.3) items per individual fish found in Crenimugil seheli. Figures 6 and 7 shows the images of microplastic particles found using two different types of microscopes.

Figure 5 Microplastic classification by morphological types in Crenimugil seheli, Atule mate, Sardinella fimbriata and Rastrelliger brachysoma gut samples from two fishing locations.

Figure 6 Images of different microplastic morphological types found in fish guts by using DM4 (1000x) USB digital electronic microscope.

Particles identified as (A) fibre (B) fragment (C) pellet (D) film.

The SEM-EDX image (Fig. 8) shows the morphology of the microplastics isolated from the fish guts. EDX analysis showed the presence of carbon, chlorine, iron, sodium, aluminium, calcium, silicon and oxygen in the sample particles. The elements found on the surface of the samples support the identification of the samples as microplastics.

Figure 7 Images of different microplastic morphological types found in fish guts by using MDSI-40X dissecting stereomicroscope.

Particles identified as (A) pellet (B) film (C) fragment (D) fibre.

Figure 8 The SEM-EDX imaging and spectrum of microplastic studied.

Particles identified as (A) fibre (B) film (C) fibre.

Discussion

Chemical characteristics of microplastic by EDX

Each analysed microplastic has different percentages of elements that characterize the material (Gniadek & Dabrowska, 2019). In the present study, the SEM-EDX analysis provides qualitative and quantitative results on the composition of the samples observed. Plastics commonly consist of elements such as carbon, hydrogen, nitrogen, oxygen, chlorine and sulphur. The quantitative analysis shows the morphological observation of the microplastic being studied. The microplastics observed in this research are marine debris that has undergone physical, chemical and biological weathering (Wang et al., 2017). Thus, the SEM-EDX analysis can represent any form of degradation undergone by the microplastic.

Microplastics are classified into different types based on their morphological characteristics, i.e., fibre, pellet, film and fragment. The SEM-EDX analysis of the samples in this study identified them as (a) fibre, (b) film and (c) fibre. The common observation of all microplastic observed under SEM-EDX was the degradation of the microplastic. The surface of the microplastic was rough with cracked and porous surfaces. Degradation of the microplastics could occur due to weathering, microbial action and chemical actions (Shahnawaz, Sangale & Ade, 2019).

The greatest percentage of elements found in the EDX analysis was potassium (K). The occurrence of K indicates the chemical deposition of KOH which was used in the digestion process, as described above (O’Donovan et al., 2018). Moreover, the presence of carbon (C) distinguishes the element as a plastic, as most of the plastics produced are primarily made up of C. The highest component of samples (a) and (b) are K, C and oxygen (O). Sample (c) is slightly different, comprised predominantly of C, K and calcium (Ca). The presence of C and O in samples (a) and (c) indicates that the microplastic is typically a polyethylene terephthalate (PET) (Kohutiar, 2020 #132). PET is a commonly used plastic material in the textile industry (Wang et al., 2017). The structure type of microplastics observed in samples (a) and (c) is fibre which is basically made up of PET. The presence of Ca also indicates the filler material that is used in the production of original plastics, which is the source of the microplastics observed in this research. Calcium carbonate (CaCO3) is the most popular filler material used in plastic compounds such as polyethylene where it reduces surface energy, enhancing the surface gloss and opacity (Elleithy et al., 2010).

Microplastic ingestion by marine fish

The present study contributes to the knowledge base on the presence of microplastics within the human food chain by providing data on the ingestion of microplastics by four commercially available fish species in the Peninsular Malaysia seawater of the Strait of Malacca. A remarkable and significant finding of this study was 100% of the individual fish gut samples (FO =100%) were found to contain microplastics. This FO is remarkably high compared to a similar study conducted in Malaysia on plastic ingestion among commercial fish (collected from the fish market in Sri Kembangan, Malaysia), where researchers found an average FO of 26% (Karbalaei et al., 2019). A similar study conducted in Indonesia and the United States (US) showed a plastic uptake of 28% and 9%, respectively. These results may not be surprising given that the US (20th), Malaysia (8th), and Indonesia (2nd) are among the top 20 countries mismanaging plastic waste (Jambeck et al., 2015). Thus, the results of the present study provide leading evidence of microplastic contamination in commercial marine fish guts from the Northwest Peninsular Malaysia seawater. This also indicates that marine fish in the studied area are exposed to and interact with microplastics in the marine environment. A recent dissertation by Hastuti, Lumbanbatu & Wardiatno (2019) showed similar results where 97.13% of FO in the sample fish guts were contaminated by microplastic. Similarly, Jabeen et al. (2017) found 100% of FO in marine fish using µFTIR and chemical digestion in the Yangtze estuary, China.

The Strait of Malacca is known for its high anthropocentric activity, and it is expected that greater plastic concentrations will occur, however, no research was available previously and the present study aimed to fill this knowledge gap. It is known that Southeast Asia is one of the worst plastic polluters (Jambeck et al., 2015). Considering that there is a paucity of research in Asian seawaters, which is emphasized by the review of Markic et al. (2020), there could well be an unknowingly high FO of microplastics in this region. The available studies in Southeast Asia found high FO (100%) for Crenimugil seheli and Sardinella fimbriata (Hastuti, Lumbanbatu & Wardiatno, 2019), 33% for Rastrelliger brachysoma (Azad et al., 2018) and 11.4% for Atule mate (Klangnurak & Chunniyom, 2020). However, small sample sizes were used, increasing the margin of error. Acknowledging these shortcomings, the magnitude of marine plastic ingestion in Southeast Asia could well be underestimated, possibly inducing vast exposure on its citizens with unknown health effects. However, further research is needed to validate this statement.

The comparison of microplastic data with other studies still has a large research gap due to differences across sample size, fish species, temporal and spatial scale as well as sampling methods. Despite that, the results of this study support the growing literature documenting microplastic ingestion by fish under natural conditions (Karbalaei et al., 2019; Akhbarizadeh, Moore & Keshavarzi, 2018; Barboza, Vieira & Guilhermino, 2018b; Neves et al., 2015; Nobr et al., 2015). These studies reveal that microplastic ingestion by marine fish from different species and feeding habitats is common nowadays (Lusher, Mchugh & Thompson, 2013).

The main type of microplastic found in the fish sample guts in the present study was fragment (49.5%), which is another remarkable finding in this study. This result supports the studies of Eriksen et al. (2014) which found that this type of isolated microplastic was most abundant. The results also indicate the widespread problem of micro-fragments in the marine ecosystem. Percentages of fibre (41.9%) were similar to fragment and ranked as the second-highest microplastic type found in this study. The result of this study was partly complemented by the result of other studies by Boerger et al. (2010), Lusher, Mchugh & Thompson (2013) and Pazos et al. (2017), which found that fibres were the predominant microplastic, indicating the high probability of fish ingesting microfibres. Together, these findings suggest that the difference in microplastics morphological types, sizes and polymers found in organisms are likely caused by different strategies of waste management, contamination sources and sampling locations.

Variation of microplastic ingestion among the four marine fish species

In this study, microplastics were present in all the sampled fish guts. Microplastic ingestion numbers varied among the four different species of commercial marine fish from two different locations. The overall finding of this study suggests bony commercial marine fish species from the Northwest Peninsular Malaysia seawater are ingesting significant quantities of microplastic particles. However, there might be risks of underestimation (loss of microplastics during sampling or lab processing) or overestimation (net feeding, background and airborne contamination) of microplastics, which could lead to misrepresentation of the data (Nadal, Alomar & Deudero, 2016; Rummel et al., 2016). Despite these risks, the sample blanks of each commercial species in this study were identified with negligible contamination levels (range 1–2 particles with mean = 1.25 particles per blank), indicating minimal airborne contamination during the processing of the fish gut samples.

According to Froese & Pauly (2000), all four species in this study (Atule mate, Crenimugil seheli, Sardinella fimbriata and Rastrelliger brachysoma) are pelagic fish with a habitat range between shallow marine and brackish water. They are commercial marine fish that are commonly found in Malaysia’s wet markets and popular as daily meals among Malaysians. A previous study by Lusher, Mchugh & Thompson (2013) reported the presence of microplastics in pelagic fish species in the English Channel. The difference in microplastic ingestion (occurrence in fish guts) in the four species in the present study might be influenced by several factors such as the prevalence of microplastic found in two different fishing locations, fish feeding behaviour and strategies, ecological range and population density (Liboiron et al., 2016). Pelagic-feeding fish are more vulnerable to microplastic particle ingestion compared to demersal feeders due to the characteristics of microplastic polymers (high buoyancy, low density) which are abundant and highly dispersed on the marine surface (HHidalgo-ruz et al., 2012). Microplastic ingestion by marine fish at a particular location may not necessarily be from local sources, due to marine currents which play a vital role in transporting microplastics over very long distances (Nadal, Alomar & Deudero, 2016).

In the present study, Sardinella fimbriata was recorded with the highest amount of microplastic ingestion, with a total number of 117 for all samples or 6.5(±4.3) items per individual fish. These results are supported by a recent study by Hastuti, Lumbanbatu & Wardiatno (2019) in which the highest microplastic numbers were found in Sardinella fimbriata with 20 ± 8 particles per individual fish. Sardinella fimbriata are filter feeders, feeding on planktonic organisms; they inhabit the marine pelagic zone and filter water while they are feeding (Hastuti, Lumbanbatu & Wardiatno, 2019). Fish with filter-feeding behaviour are more susceptible to microplastic ingestion compared to other marine fish as they are generalists in term of feeding behaviour (Rummel et al., 2016). Also, the high microplastic ingestion of this species might due to the feeding habitat at the shallow pelagic marine zone. Floating and buoyant microplastic are highly bioavailable for this species as they are extremely small and similar to natural prey which can be found within the water column’s plankton (Lima, Costa & Barletta, 2014). The finding that Sardinella fimbriata ingests a significant quantity of microplastics suggests the potential of this species as an indicator for further studies on microplastic ingestion in marine fish. The present study also suggests the possibility that plankton- and pelagic-feeding fish such as Sardinella fimbriata could be a major sink species for the floating microplastics in the marine waters of Malaysia.

Crenimugil seheli was shown to ingest the lowest amount of microplastics with a total number of 90 for all samples or an average number of 5.0(±3.7) items per individual fish. Although this species is a pelagic feeder like Atule mate, Sardinella fimbriata and Rastrelliger brachysoma, the Crenimugil seheli was fished at a different site to the other three sample species of this study. The fishing location of Crenimugil seheli was nearby the coast of Teluk Bahang while the Atule mate, Sardinella fimbriata and Rastrelliger brachysoma’s fishing location was nearby the coast of Penaga. This suggests that the marine environment of the fishing location at Teluk Bahang (located at the Penang National Park) might be less polluted by microplastic debris compared to the fishing location of Penaga. When considering feeding behaviour, Crenimugil seheli is herbivorous while Sardinella fimbriata is carnivorous (Hastuti, Lumbanbatu & Wardiatno, 2019). According to Froese & Pauly (2000), Atule mate and Rastrelliger brachysoma feed mostly on small crustaceans, planktonic invertebrates and microzooplankton, and can be classified as carnivorous. Hastuti, Lumbanbatu & Wardiatno (2019) found that herbivorous fish have a lower average microplastic number than carnivorous fish, indicating that carnivorous fish might accumulate more microplastic particles. This is due to trophic transfer during which the microplastics transfer from prey to carnivorous fish or are mistakenly ingested as natural prey (Lusher, Mchugh & Thompson, 2013; Markic et al., 2018, Boerger et al., 2010). Our results support this finding, showing that Crenimugil seheli, as a herbivorous fish, ingested less microplastics compared to the other three carnivorous fish species. However, robust evidence regarding the influence of trophic transfer on the microplastic ingestion rate in marine fish is still lacking. The multispecies survey of Güven et al. (2017) found a contradictory result in which there was no correlation of microplastic ingestion with fish biological parameters and the fish species’ trophic level. Güven et al. (2017) also proposed that only the habitat types or ranges (benthic or pelagic fish) will influence microplastic ingestion by marine fish. Thus, many different factors need to be taken into consideration when comparing microplastic ingestion across different fish species, such as the differences in biological and physiological characteristics, ecological range, feeding behaviour and food retention time in the gut (Grigorakis, Mason & Drouillard, 2017; Ory et al., 2018; Hastuti, Lumbanbatu & Wardiatno, 2019).

Negative impacts of microplastics ingestion

Microplastic pollution also negatively affects a wide range of taxonomic groups in the marine biota including zooplankton, sea urchins, sea turtles and corals (Browne et al., 2008). The finding of microplastic ingestion in marine fish could be a starting point for further investigations on the impacts and toxicity of microplastics to fish health and the further risks of exposure to the marine biota via trophic transfer. The consumption of fish and other seafood contaminated with microplastics could potentially impact human health, although this assumption is not proven. More dynamic studies regarding microplastics in marine organisms are needed to have a better understanding of the impacts of microplastics on fish, the whole marine biota and humans.

Human health implications

Numerous pathways of microplastic exposure in humans exist, however, knowledge on the human health effects is largely unknown (Wright & Kelly, 2017). The human health implications research field is in the early stages (Koelmans et al., 2017). Nevertheless, there are sundry commercial fish species bringing seafood safety into question. Additionally, a recent study found that microplastics are capable of penetrating root systems and contaminating fruit and vegetables (Oliveri Conti et al., 2020). The food we consume and the omnipresence of microplastics increasingly affect the very foundation of our lives. Microplastics are increasingly revealed as a potential public health concern.

According to a literature review by Barboza et al. (2018a), there is no consensus on microplastics size that can transfer from the gut cavity to the lymph and circulatory system of humans. Currently, it is speculated that microplastics bigger than 150 µm cannot be absorbed in the human body and will be egested (Fig. 9). Smaller particles would be able to penetrate human organs. The smaller their sizes, the higher the ability.

Figure 9 Particle sizes and transferability. Data source: Lusher et al. (2017).

Plastics are expected to be toxic and could disrupt human hormones, but the real toxic effects are mostly unknown (Wright & Kelly, 2017). Nevertheless, we know from the literature that plastics toxicity increases with dose and smaller particle sizes (MATTSSON et al., 2017). Microplastics have been found in the edible part of the fish (Abbasi et al., 2018; Karami et al., 2017b). In two species, the fish fillet contained higher plastic loads than the gastrointestinal tract. De-gutting before cooking is common practice for the four fish species (Atule mate, Crenimugil seheli, Sardinella fimbriata, and Rastrelliger brachysoma) in this study. Thus, the guts of the four species are generally not consumed by humans, and only the fish fillet is consumed. There might be an accumulation of microplastics in consumer food, but what this means for human health remains a challenge based on limited data to date. Exposure to plastics and additives in our society are ubiquitous, but the specific impacts on human health are not entirely understood, and many knowledge gaps prevail.

Conclusions

In conclusion, the results of this study indicate that microplastic ingestion by commercial fish from the Northwest Peninsular Malaysia seawater does occur, suggesting a potential route of microplastic exposure to humans. The most abundant microplastics found in this study were fragments and fibres, with chemical microplastic composition was confirmed by SEM-EDX. The fish gut are often discarded before human consumption, thus further investigation of microplastic contamination of edible and commercial fish tissue is recommended to assess potential microplastic pollution in human food. Thus, further research is needed for a deeper understanding and risk assessment of marine food safety and security in Malaysia.

Supplemental Information

Supplemental Information 1 Biometrics parameter for fish

Click here for additional data file.

Supplemental Information 2 Result of microplastic found in fish gut

Click here for additional data file.

Supplemental Information 3 SEM EDX analysis

Click here for additional data file.

We would like to thank the staff of the Centre for Marine and Coastal Studies (CEMACS) and the School of Biological Sciences, Universiti Sains Malaysia for infrastructure and technical support. Thank you also to the National Marine Environmental Monitoring Centre (NMEMC), China, for knowledge sharing regarding this research.

Additional Information and Declarations

Competing Interests

Author Contributions

Data Availability

The authors declare there are no competing interests.

Yuen Hwei Foo conceived and designed the experiments, performed the experiments, prepared figures and/or tables, authored or reviewed drafts of the paper, and approved the final draft.

Sharnietha Ratnam and Masthurah Abdullah performed the experiments, prepared figures and/or tables, authored or reviewed drafts of the paper, and approved the final draft.

Er Vin Lim performed the experiments, authored or reviewed drafts of the paper, and approved the final draft.

Vincent J. Molenaar analyzed the data, prepared figures and/or tables, authored or reviewed drafts of the paper, and approved the final draft.

Aileen Tan Shau Hwai, Shoufeng Zhang and Hongjun Li analyzed the data, authored or reviewed drafts of the paper, and approved the final draft.

Norlaila Binti Mohd Zanuri conceived and designed the experiments, prepared figures and/or tables, authored or reviewed drafts of the paper, and approved the final draft.

The following information was supplied regarding data availability:

The raw data measurements are available in the Supplementary Files.

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
