# Peer review of "Microplastic ingestion by commercial marine fish from the seawater of Northwest Peninsular Malaysia"

_PeerJ, doi:10.7717/peerj.13181_

## Round 0.1 · original submission · Major Revisions

Dear authors
After reading the reviewers' suggestions, I decided to suggest the publication after major revisions. Please read the comments carefully. The English grammar and Statistics deserve attention.
Regards,

Reviewer 1 ·

Basic reporting

The article "Microplastic ingestion by commercial marine fish from the seawater of Northwest Peninsular Malaysia" written by Foo, Zanuri et. Al, shows an interest in microplastic consumption by four different commercial fishes in Malaysia.

Microplastic ingestion in marine organisms is well documented around the world but new results are always welcome since this type of pollution is still increasing and solutions for the problem are far from becoming a worldwide reality.

Although the paper abstract and introduction are relevant and well documented as well as figures and data, the quality of the English language must be improved. Among several English difficulties I could point out orthographic problems (such as “tophic” instead of trophic at line 107), sentence repetition (lines 347-349 and 350 – 352) and long and overloaded phrases. This manuscript should be reviewed for its English prior to publication.

The manuscript structure complies with the journal norms and the raw data was supplied.

Experimental design

The research fills the aim and scope of the journal, since it tries to understand the microplastic ingestion by commercial fish in Malaysia. The authors aimed to elucidate the level of microplastic contamination in fishes, which was explained very well along the manuscript. It obviously fills an important gap in the Asian microplastic knowledge since literature lacks information on sea and marine organisms’ contamination from the area.

The experiment was designed with no negative control. This can be understood since there is a lack of microplastic pollution dispersion information in the region making it impossible to the authors to compare the commercial fishes microplastic ingestion to fishes’ contamination from cleaner regions.

The number of individuals analyzed was good, 72 in total, which would be enough to perform a robust statistical analysis which was not shown. This could improve results since the authors would be able to compare the number of microplastics ingested from the two different fishing areas. These results would allow authors not only hypothesize about different microplastic pollution degrees in the two different sites but to have evidences that this may be occurring.

The fish manipulation and digestion protocols are well documented allowing others to replicate the experiment with other species and areas so results could be compared.

Work with microplastics identification should be done with rigorous protocols to avoid external contaminations. Authors indicates they were aware of the problem, but even though precautions were made, the control digestion experiment detected some microplastic presence, which could be something to explain better, or to insert in their data analysis.

Validity of the findings

Results are well described, and the raw data was attached to eventual observations. The total number of individuals is good enough to perform statistical analysis which I was not able to identify in the manuscript. The figures are well labeled but in the manuscript the authors go from figure 5 (line 278) to figure 8 (line 289). It seems to me that figure 6 and 7 are not referenced in the text thus it shows the microplastic particles on microscopical views which is interesting since until there, there was no mention of microplastic observation.

The SEM-EDX results could be improved and references should be annotated (from line 313 to 318 the authors affirm that presence of carbon and oxygen, potassium and calcium are indication of plastic but in any moment they indicates a literature to substantiate findings).

The conclusions are well stated and limited to the findings. Humans' microplastic contamination possibility and effects are speculated and identified as something remote since there are not enough studies to confirm it, even if microplastic may be found in consumable fish parts.

Additional comments

The manuscript is interesting and well designed but it lacks english lanquage quality that should be assessed before publication, The results have potential to initiate other experiments and new insights about plastic contamination in the area of study which is always something good to the future of your research.
The difficulties of analyzing environemental sampled microplastics are inherent to the ubiquity of ther presence, but your work, using different analysis methods was able to not only identify microplastics but to try to indicate what type of plastic there were in the gut of commercial fishes.
I guess that with robust statistical analysis and a better english language, the article wil be more pleasant to read.

Reviewer 2 ·

Basic reporting

- Language and grammar issues: The English language needs be improved to ensure that an international audience can clearly understand your text. Some examples where the language could be improved along the introduction– the current phrasing makes comprehension difficult. I suggest you have a colleague who is proficient in English and familiar with the subject matter review your manuscript, or contact a professional editing service.

- Introduction, background & discussion: Can still be improved and more aligned

- Literature was relatively well referenced & relevant ☺

- Figures are relevant and images are of good quality. Labeling & description can be improved.

- Raw data was supplied

- No hypothesis was tested or clear research question attempted to be answered. It is important to expand upon the knowledge gap being filled. To me it seems that the major knowledge gap filled was in regards to contributing to data collected in Malaysia.

Experimental design

- Sampling design was intrinsically confounded by species analysed in the different sampling location/ dates. Crenimugil seheli (collected on November 27th, 2019 at the 
location 1) was compared to Atule mate, Sardinella 
fimbriata and Rastrelliger brachysoma (collected on 
December 3rd, 2019, on location b). You nicely discussed the possible issues related to differences in the sampling dates, locations and other factors such as feeding behaviour (lines 405-414 and 433-466).

- Methods were described with sufficient detail & information to be replicated

- Results on the average abundance (numbers in each species and numbers in individual fish, figures 3 and 4) indicated that amounts of plastics were relatively similar among the three species collected from location 2, although slightly greater to Sardinella. It seems that if you run a statistical test (e.g. analyses of variance), you would find no significant differences in the abundance of plastics among the three sps collected from location 2. After having results from a statistical test, the emphasis you gave Sardinella having the highest abundance should be reviewed on results, discussion and conclusion.

Validity of the findings

The data provided is a valid and adds information about the presence, abundance and types of micro plastics found in the guts of commercial fish .

However, discussion and conclusions were not clearly connected to a research question and conclusions were not limited to those supported by results from this MS.

Additional comments

General comments:
There is some interesting information in the MS, albeit not unexpected, but it needs important conceptual change and revision to make it a useful contribution to our understanding of the microplastics ingestion by pelagic fish more generally and to its possible transference to humans.
I have made some comments using sticky notes in the PDF and have elaborated some key points below.

Detailed comments

Abstract
Line 62: Replace ‘fishes’ by ‘fish’. The plural of fish is actually fish … awkward things of the English language. Check all the remaining text…

Introduction

Remarks:
Were there any hypothesis being tested?

The significance of the work needs to be more clearly stated during the introduction. Although microplastics ingestion by fish have been previously reported around the world, there is relatively small information from Malaysian’s commercial coastal species? Are there any other significant contributions? Explain it please

Lines 75-78: Please revise the concepts of degradation and biodegradation and the link between ideas was not clear.
I suggest rephrasing as: Plastics are extremely resistant to biodegradation. They, however, degrade into smaller particles over time when predisposed to several natural factors, such as sunlight and wave action (Wang et al., 2016).
Delete: the bit “and therefore rapidly accumulate in marine ecosystems that could be very 
harmful especially for the marine environment”. (Does not link with previous idea and, its not because they degrade over time that they accumulate in marine ecosystems… 


Lines 81-82: The increase reach of plastics in natural environments is also due to its mismanagement. I suggest rephrase it as:
“Plastics will
 increasingly reach all areas of our environment inherent to the growth of plastic production (Azoulay et al., 2019) and its mismanagement during production, distribution, use and final disposal.

Line 91: Check definition of primary microplastics. Not all of them are produced as pellets. Primary microplastics are I intentionally produced by industries within this size range, for direct applications (i.e., medicine and cosmetics). Primary microplastics include industrial ‘scrubbers’ used to blast clean surfaces, plastic powders used in moulding, micro-beads in cosmetic formulation, and plastic nanoparticles used in a variety of industrial pro- cesses. In addition, spherical or cylindrical virgin resin pellets… (see “GESAMP (2015). “Sources, fate and effects of microplastics in the marine environment: a global assessment” (Kershaw, P. J., ed.). IMO/FAO/UNESCO-IOC/UNIDO/WMO/IAEA/UN/UNEP/UNDP Joint Group of Experts on the Scientific Aspects of Marine Environmental Protection). Rep. Stud. GESAMP No. 90, 96 p.
Line 99: Primary microplastics can also be ingested. Suggest rephrase as: ‘Primary and secondary microplastics have the potential to be ingested …’
Lines 115 – 128: that hole part belongs to the methods


Materials and Methods

Remarks: It was not clear if you used an already available standardized protocol OR if you a proposing a new protocol. From your text and references it seems that you’ve used a combination of previous protocols from Lusher et al., 2017 and Dehaut et al., 2016.
 If that’s correct, why did you adapt protocols?

Lines 136-140: Your sampling design is intrinsically confounded by species analysed in the different sampling dates. It’s not clear why you collected Crenimugil seheli 
on the first sampling date and other three different species on the second…This rational or problems associated with your sampling strategy need to be explained.


Results

Remarks:
Do you have data about the actual sizes of microplastics found in your samples? This would greatly contribute to a possible speculation about transference to humans via consumption of fish

Review your statements about greatest abundance of microplastics in Sardinella

Lines 272 -277: Needs to be re-structured / rephrased
Suggestion would be to say “Meanwhile, the abundance of micropalstics was greater on three species collected at location 2. Relatively similar abundances were among the three species collected from location 2, although slightly greater to Sardinella.”


Discussion

Line 347: Start a new paragraph

Lines 350-351: Delete, repetition of previous phrase.

Lines 364-367: Concentrate your discussion around your research focus, ingestion by fish under natural conditions.

Lines 367-370: “The result of 
 this study coincides with the study of Browne et al. (2010) as well, which stated that the 
arising risks of microplastic ingestion by marine animals indicates microplastic is more 
abundant than large plastic debris in the marine environment now.” Suggest to remove that bit or rephrase it …. Browne et al. (2010) evaluated spatial patterns of macro and microplastics fragments. Their results showed that microplastic accounted for 65% of debris recorded. They mentioned that when macroplastics fragments into smaller pieces, the potential for ingestion and accumulation within the tissues of animals increases. They don’t stated that arising risks of microplastic ingestion by marine animals indicates microplastic is more 
abundant than large plastic debris in the marine environment now…
Lines 419-420: “Sardinella fimbriata is 
planktonic organisms and filter feeders”. That needs to be rephrased… Did you mean to say “Sardinella fimbriata are filter feeders and feed on planktonic organisms?
Lines 478 – 503: Possible impacts on human’s health can be related to the amounts and sizes of microplastics ingested, among other factors. Having this in mind, if you are going to speculate about transference to humans, it would be important to explain if guts of the four species are normally ingested as human’s food or if only the fish filed is consumed. Also, if possible please provide information about the sizes of the microplastics found on your samples and state the sizes of MP found by other authors (e.g. Abbasi et al., 2018)

Conclusions:

Remarks: needs to be linked to original research question & limited to supporting results.
See detailed comments in the text.

Annotated reviews are not available for download in order to protect the identity of reviewers who chose to remain anonymous.

---

## Round 0.2 · Minor Revisions

Dear authors

Reviewers considered that the manuscript still needs minor revisions. I will consider it for publication after you consider their comments.

Reviewer 1 ·

Basic reporting

changes were made accordingly to reviewers suggestions

Experimental design

changes were made accordingly to reviewers suggestions

Validity of the findings

changes were made accordingly to reviewers suggestions

Additional comments

To explain the tested hypothesis, authors made a bullet list in Material and Methods. It would be better to incorporate the hypothesis at the end of the introduction as a discursive text.

---

## Round 0.3 · accepted · Accept

Dear Authors
I am very happy to announce that your paper can be accepted for publication.
Thank you

Reviewer 1 ·

Basic reporting

The authors revised the text according to prior reviews.

Experimental design

The authors revised the text according to prior reviews.

Validity of the findings

The authors revised the text according to prior reviews.